# Immunophenotypic Landscape and Prognosis of Diffuse Large B-Cell Lymphoma with MYC/BCL2 Double Expression: An Analysis of A Prospectively Immunoprofiled Cohort

**DOI:** 10.3390/cancers12113305

**Published:** 2020-11-09

**Authors:** Bogyeong Han, Sehui Kim, Jiwon Koh, Jeemin Yim, Cheol Lee, Dae Seog Heo, Tae Min Kim, Jin Ho Paik, Yoon Kyung Jeon

**Affiliations:** 1Department of Pathology, Seoul National University Hospital, Seoul National University College of Medicine, Seoul 03080, Korea; vokyong@snu.ac.kr (B.H.); greenghost@snu.ac.kr (S.K.); jiwonsophia@snuh.org (J.K.); dayimmer@snu.ac.kr (J.Y.); fe98134@snu.ac.kr (C.L.); 2Cancer Research Institute, Seoul National University, Seoul 03080, Korea; heo1013@snu.ac.kr (D.S.H.); gabriel9@snu.ac.kr (T.M.K.); 3Department of Internal Medicine, Seoul National University Hospital, Seoul National University College of Medicine, Seoul 03080, Korea; 4Department of Pathology, Seoul National University Bundang Hospital, Seongnam-si 13620, Korea; paikjh@snu.ac.kr

**Keywords:** diffuse large B-cell lymphoma, high-grade B-cell lymphoma, MYC, BCL2, double expression, prognosis, lactate dehydrogenase

## Abstract

**Simple Summary:**

Diffuse large B-cell lymphoma (DLBCL) with MYC/BCL2 double-expression (DE), a recently proposed poor prognostic group, can be easily identified by immunohistochemistry in routine clinical practice. However, clinical outcomes of DE-DLBCL patients vary immensely after R-CHOP immunochemotherapy and prognostic impact of MYC/BCL2-DE was conflicting according to the cell-of-origin, i.e., between germinal center-B-cell (GCB)- and non-GCB-DLBCLs. This implies the heterogeneity within DE-DLBCLs and emphasizes a need for proper risk stratification to select the patients who require more intensive therapy. By analyzing a prospectively immunoprofiled cohort of consecutively diagnosed DLBCL patients, we confirmed the poor prognostic value of MYC/BCL2-DE in DLBCL patients treated with R-CHOP irrespective of the cell-of-origin and international prognostic index. DE-DLBCLs with a concurrent risk factor, especially, elevated serum lactate dehydrogenase (LDH), had the worst survival and DE-DLBCL patients with normal LDH had clinical outcomes similar to those of non-DE-DLBCL patients. Risk stratification of DE-DLBCL based on serum LDH may guide clinical decision-making for DE-DLBCL patients.

**Abstract:**

Diffuse large B-cell lymphoma (DLBCL) patients with MYC/BCL2 double expression (DE) show poor prognosis and their clinical outcomes after R-CHOP therapy vary immensely. We investigated the prognostic value of DE in aggressive B-cell lymphoma patients (*n* = 461), including those with DLBCL (*n* = 417) and high-grade B-cell lymphoma (HGBL; *n* = 44), in a prospectively immunoprofiled cohort. DE was observed in 27.8% of DLBCLs and 43.2% of HGBLs (*p* = 0.058). DE-DLBCL patients were older (*p* = 0.040) and more frequently exhibited elevated serum LDH levels (*p* = 0.002), higher international prognostic index (IPI; *p* = 0.042), non-germinal-center B-cell phenotype (*p* < 0.001), and poor response to therapy (*p* = 0.042) compared to non-DE-DLBCL patients. In R-CHOP-treated DLBCL patients, DE status predicted poor PFS and OS independently of IPI (*p* < 0.001 for both). Additionally, in DE-DLBCL patients, older age (>60 years; *p* = 0.017), involvement of ≥2 extranodal sites (*p* = 0.021), bone marrow involvement (*p* = 0.001), high IPI (*p* = 0.017), CD10 expression (*p* = 0.006), poor performance status (*p* = 0.028), and elevated LDH levels (*p* < 0.001) were significantly associated with poor OS. Notably, DE-DLBCL patients with normal LDH levels exhibited similar PFS and OS to those of patients with non-DE-DLBCL. Our findings suggest that MYC/BCL2 DE predicts poor prognosis in DLBCL. Risk stratification of DE-DLBCL patients based on LDH levels may guide clinical decision-making for DE-DLBCL patients.

## 1. Introduction

Significant progress in the classification of aggressive B-cell lymphoma has been made in the last decade [1,2]. Common entities include diffuse large B-cell lymphoma (DLBCL), Burkitt lymphoma (BL), and high-grade B-cell lymphoma (HGBL) [1,2]. Except for BL, DLBCLs and HGBLs are characterized by heterogeneous pathological, genetic, and clinical features, increasing the need for more accurate disease clarification and prognostication. HGBLs harboring MYC and BCL2 or BCL6 translocation, commonly known as double-hit (DH) lymphoma, are frequently refractory to therapy, leading to early relapse and poor patient prognosis [3,4,5]. Evidence from retrospective studies shows that DH lymphoma patients receiving intensive treatment exhibit a better clinical outcome than patients treated with rituximab plus cyclophosphamide, doxorubicin, vincristine, and prednisone (R-CHOP) [6,7,8]. Thus, DH lymphoma should be considered as a distinct entity according to the revised 4th World Health Organization (WHO) classification of lymphoma [9]. 

DH lymphomas account for 1–8% (~5% on average, 2% in Korea) of de novo DLBCLs [3,5,10,11,12,13,14,15], and 30–50% of HGBLs [9,16], and typically display a germinal center B-cell (GCB) phenotype [12,13,17]. Follicular lymphomas, which could progress to high-grade lymphoma with DH status, account for 5.3% of lymphomas in Korea [18], while BCL2 translocations are observed in 3.4% of Korean DLBCL patients, both of which are considerably lower than frequencies observed in Western populations [15]. Morphologically, the majority (50–69%) of DH lymphomas resemble DLBCL and others mimic BL [9,19,20]. 

Aggressive B-cell lymphomas gain MYC and BCL2 alterations through mechanisms other than gene translocations, and 18–44% of DLBCLs have been reported to express MYC and BCL2 concurrently [10,11,12,13,14]. In contrast to DH lymphoma, DLBCLs with MYC/BCL2 double expression (DE-DLBCL) frequently exhibit activated B-cell-like (ABC) or non-GCB phenotypes [10,11,12,13]. Notably, patients with DE-DLBCL show unfavorable prognosis [1,6,12,13,21]. In contrast to DH lymphomas, response to R-CHOP varies significantly among DE-DLBCLs, and there is no reliable prognostic factor within this group warranting more aggressive treatments [7]. Therefore, the development of novel approaches to stratify DE-DLBCL patients into high-risk and low-risk groups are necessary.

Diverse genetic and immunohistochemical methods have been developed to determine the cell-of-origin (COO) of DLBCL. It has become evident that the COO in DLBCL varies considerably among populations [22]. The GCB phenotype is observed in approximately 60% of DLBCLs in Western countries, whereas the non-GCB/ABC phenotype is seen in 60% of Korean DLBCL patients [22,23,24]. Although the prognostic implications of the COO remain a matter of debate [23,25,26], determining the COO is recommended for all DLBCL patients [1,9]. 

In this study, we prospectively analyzed the immunophenotypic landscape, DE status, and COO of 461 consecutively diagnosed DLBCL and HGBL patients using immunohistochemical analysis. We also investigated the clinicopathological implications of DE status in DLBCL and HGBL patients and the potential of the risk stratification strategy for DE-DLBCL patients treated with R-CHOP. 

## 2. Results

### 2.1. Clinicopathological Characteristics of Patients 

We enrolled 461 aggressive B-cell lymphoma patients, including 417 (90.5%) DLBCL patients and 44 (9.5%) HGBL patients. The clinicopathological characteristics of the patients are summarized in Table 1 and Figure 1. Patients with DLBCL were significantly older than those with HGBL, with a median age of 62 years versus 56 years, respectively (*p* = 0.002). Bulky disease was more frequently observed in patients with HGBL (37.2%) than in those with DLBCL (8.3%) (*p* < 0.001). There was no statistically significant difference in other clinical features between DLBCL and HGBL patients. Approximately 98% of DLBCL patients were treated with the R-CHOP regimen, whereas R-CHOP was administered to 69% of HGBL patients (*p* < 0.001). Other treatment regimens included CHOP (*n* = 1), R-EPOCH (*n* = 1), BVP (*n* = 1), R-HyperCVAD (*n* = 1), R-CVP (*n* = 1), CDP (*n* = 1), and Prednisone (*n* = 1) in DLBCL patients, and R-EPOCH (*n* = 1), EPOCH (*n* = 1), R-HyperCVAD (*n* = 4), and R-dmCODOX (*n* = 7) in HGBL patients. These findings indicate that HGBL patients were more frequently treated with aggressive regimens compared with DLBCL patients. However, there was no significant difference in the rate of complete response (CR) between DLBCL and HGBL patients.

### 2.2. Immunophenotypic Landscape of DLBCL and HGBL 

The immunophenotypic landscape of aggressive B-cell lymphoma is summarized in Table 1 and illustrated in Figure 1. Representative pathological images of a DE-DLBCL patient are displayed in Appendix A. CD10 and BCL6 were more frequently expressed in HGBLs than in DLBCLs (*p* < 0.001 and *p* = 0.01, respectively). Furthermore, COO distribution significantly differed between DLBCLs and HGBLs (*p* < 0.001); 62.7% of DLBCLs showed the non-GCB phenotype, whereas 69.2% of HGBLs exhibited the GCB phenotype. Although MYC expression was more common in HGBLs (74.4%) compared with DLBCLs (42.2%) (*p* < 0.001), the frequencies of DE-positive cases did not differ significantly between HGBLs and DLBCL (43.2% vs. 27.8%; *p* = 0.058). Overall, most of DLBCLs (39%) belonged to the non-GCB-/non-DE subtype, followed by GCB-/non-DE (33%), non-GCB/DE (23%), and GCB-/DE (5%) subtypes.

### 2.3. Relationship between MYC/BCL2 DE Status and Clinicopathological Features

The clinicopathological features of DLBCL patients with DE and non-DE status were compared and are summarized in Table 2. DE-DLBCL patients were slightly older than non-DE-DLBCL patients (median age 64 vs. 62 years; *p* = 0.040), and more frequently showed an elevated serum lactate dehydrogenase (LDH) level (65.7% vs. 47.3%; *p* = 0.002). COO also differed significantly according to the DE status, with the non-GCB phenotype being observed in 83.3% and 54.3% of DE-DLBCLs and non-DE-DLBCLs, respectively (*p* < 0.001; Table 2). There were no significant differences in the treatment provided to DE and non-DE-DLBCL patients. Compared with non-DE-DLBCL patients, patients with DE-DLBCL exhibited a worse response to treatment (*p* = 0.042) and higher relapse and progression rates (*p* < 0.001; Table 2). However, there was no significant difference in the clinical features of patients with HGBL according to the DE status. The non-GCB phenotype was more frequent in HGBLs with DE (6/15, 40%) than in HGBLs with non-DE status (5/21, 23.8%); however, this difference did not reach statistical significance (Figure 1).

### 2.4. Effect of DE Status on Patient Survival after Treatment with R-CHOP

There was no difference in the overall survival (OS) of patients with DLBCLs and HGBLs in the whole cohort (Appendix A) or in the sub-cohort of patients treated with R-CHOP (Appendix A). DE status was significantly associated with poor PFS and OS in DLBCL patients (*p* < 0.001 for both; Figure 2A,B) but not in HGBL patients. Of note, DE status was related to poor overall survival (OS) and progression-free survival (PFS) in both GCB and non-GCB types of DLBCL (*p* < 0.001 for all; Figure 2C–F). Hereafter, subsequent survival analyses were performed in DLBCL patients treated with R-CHOP. Univariate Cox survival analysis of DLBCL patients treated with R-CHOP demonstrated that high Ann Arbor stage, poor (≥2) Eastern Cooperative Oncology Group (ECOG) performance status (PS), elevated serum LDH, involvement of two or more extranodal sites, bone marrow (BM) involvement, high international prognostic index (IPI) score, BCL2 positivity, MYC positivity, and MYC/BCL2 DE status were significantly associated with poor PFS and OS (Appendix A). Multivariate survival analysis revealed that DE status was a significant poor prognostic factor for PFS and OS, independently of age, sex, stage, ECOG PS, serum LDH level, and the number of extranodal sites (*p* < 0.001 for PFS and *p* = 0.006 for OS), as well as independently of IPI score (*p* < 0.001 for both PFS and OS; Table 3). In addition, DE status was a significant poor prognostic factor for PFS and OS independently of IPI score in both the GCB and non-GCB DLBCL groups, respectively (Appendix A).

### 2.5. Prognostic Stratification of DE-DLBCL Patients Treated with R-CHOP 

Univariate Cox survival analysis of DE-DLBCL patients demonstrated that age, poor ECOG PS, elevated serum LDH, involvement of two or more extranodal sites, BM involvement, high IPI score, CD10 expression, and GCB phenotype were associated with poor OS (*p* < 0.05 for all; Appendix A). To further stratify DE-DLBCL patients based on their prognosis, we classified patients into three groups: non-DE-DLBCL, DE-DLBCL without a risk factor, and DE-DLBCL with a risk factor. In DE-DLBCL patients, older age, poor ECOG PS, BM involvement, involvement of two or more extranodal sites, CD10 expression, and GCB phenotype were the most significant factors predicting poor prognosis (*p* < 0.05 for all; Figure 3 and Appendix A). OS of DE-DLBCL patients who did not harbor these risk factors were between the other two groups, i.e., non-DE-DLBCL patients and DE-DLBCL patients with these risk factors (*p* < 0.05 for all; Figure 3). In contrast, DE-DLBCL patients with low IPI (0 or 1) had similar PFS and OS to those of patients with non-DE-DLBCL (Figure 3D and Appendix A). Moreover, DE-DLBCL patients with normal LDH levels exhibited similar PFS and OS to those of non-DE-DLBCL patients (Figure 3B and Appendix A). 

### 2.6. Validation of Prognostic Stratification of DE-DLBCLs

As a validation set, we used publicly available data generated by Schmitz et al [27]. Of 234 patients in the validation set, LDH levels were available for 202 patients, 49 of whom were classified as DE-DLBCL. Comparison of OS and PFS among DE-DLBCL patients with elevated LDH levels, DE-DLBCL patients with normal LDH levels, and non-DE-DLBCL patients revealed that DE-DLBCL patients with elevated LDH levels had the worst survival (Appendix A); DE-DLBCL patients with normal LDH levels and non-DE-DLBCL patients had a similar prognosis. Additional survival analyses of DE-DLBCL patients according to PS, age, and the number of involved extranodal sites revealed that poor performance and older age were significantly associated with poor OS (Appendix A). Furthermore, in DE-DLBCL patients, high IPI scores and ABC phenotype were associated with poor prognosis (Appendix A). 

## 3. Discussion

In this study, we performed immunohistochemistry (IHC) staining in whole tissue sections obtained from a prospective cohort of aggressive B-cell lymphoma at the time of initial diagnosis to assess the prognostic value of COO and DE status. We found that DE status was a poor prognostic factor in DLBCL patients independently of IPI, consistent with previous studies showing an association of DE status with poor patient prognosis [11,12,13,21]. In particular, we found that DE status was significantly associated with poor PFS and OS in both GCB and non-GCB-DLBCL patients treated with R-CHOP. There have been conflicting reports on the prognostic implication of DE status in DLBCL with different COO. Hu et al. and Green et al. demonstrated that DE status was associated with a poor prognosis in both GCB and ABC/non-GCB-DLBCLs [12,13]. In contrast, other studies have reported that, despite the association between DE status and poor prognosis in GCB-DLBCLs, DE status could not predict survival in patients with ABC/non-GCB-DLBCLs [23,25,28]. These discrepancies might be attributed to differences in the methods used for COO classification among studies, characteristics of study populations, and the retrospective or prospective nature of data collection. These findings highlight the complex relationship between COO, DE status, and DLBCL patient prognosis, which requires further investigation. 

The prognostic value of COO determined by IHC has also been controversial [26,28,29,30]. In this study, we determined the COO using IHC and found that the COO failed to predict outcomes in patients with DLBCL. Nevertheless, we observed that the non-GCB phenotype was more frequent in DE-DLBCL patients than the GCB phenotype, consistent with previous studies investigating COO using gene expression profiles, Lymph2Cx assay, or IHC [12,13,25]. These findings further support the idea that determining the COO in DLBCL might provide insight into the biology of DLBCL, though having limited value as a prognostic indicator. Hence, implementing the immunohistochemical evaluation of COO and DE status in routine clinical practice may improve therapeutic decision-making and improve treatment outcomes in DLBCL.

In this study, we found that 27.8% of DLBCLs showed MYC/BCL2 DE, which was similar to previously reported frequencies of DE-DLBCLs [10,11,12,13,14,31,32]. The vast majority of both DE and non-DE patients were treated with R-CHOP; however, CR rates were lower in DE-DLBCL patients (80.8%) than in non-DE-DLBCL patients (90.3%). The inferior response rates to R-CHOP and poor clinical outcomes of DE-DLBCL patients highlight the need for alternative therapeutic approaches for patients with DE-DLBCL. Given the significant variations in prognosis among DE-DLBCL patients, we further stratified patients based on different clinicopathological parameters. Among these clinical parameters, IPI, age, ECOG PS, number of involved extranodal sites, and BM involvement were significant prognostic factors in DE-DLBCL. Additionally, high serum LDH levels were associated with reduced PFS and OS in DE-DLBCL patients. Moreover, we found that DE-DLBCL patients with normal serum LDH levels or low IPI (0 or 1) had similar PFS and OS to those of non-DE-DLBCL patients. The significant prognostic value of serum LDH levels in DE-DLBCL was also confirmed in the validation cohort. Previous studies reported that among other clinicopathological parameters, elevated serum LDH levels had the highest prognostic value in DLBCLs [33]. Additionally, LDH levels have been associated with high tumor burden and immune suppression [34]. Thus, our findings pave the way for further prognostic stratification of DE-DLBCL patients.

Previous studies have shown that among DLBCL patients treated with R-CHOP or R-CHOP-like therapy, patients with the GCB phenotype exhibited favorable outcomes [12,26,29], contradicting our findings. We also found that DE-DLBCL patients with the GCB phenotype or CD10 expression had a worse prognosis than DE-DLBCL patients with the non-GCB phenotype or no CD10 expression (Figure 3 and Appendix A). Hans algorithm-based COO analysis in a validation cohort from SNUBH (Appendix A) confirmed that CD10 expression and GCB phenotype were associated with shorter survival in DE-DLBCL patients (Appendix A). Given that HGBL-DH/TH frequently exhibit GCB phenotypes, it is possible that *MYC/BCL2* translocations may have a stronger impact on the prognosis of DE-DLBCL patients with the GCB phenotype than those with the non-GCB phenotype. To address this issue, we retrospectively performed tissue microarray (TMA)-based fluorescent in situ hybridization (FISH) analyses in 50 selected DLBCLs with available tissue samples for TMA construction. However, many DE-DLBCL patients with the GCB phenotype were diagnosed based on biopsies, and old FFPE specimens failed to produce robust signals. In total, 28% and 16% of the tested cases for *MYC* and *BCL6*, respectively, resulted in inadequate signals, and *BCL2* FISH did not produce interpretable signals in all samples, hindering the identification of HGBL-DH/TH within this sub-cohort. Analysis of publicly available data demonstrated that DE-DLBCL patients with the ABC phenotype had a worse prognosis than those with the GCB phenotype (Appendix A), contradicting the results from our patients. This discrepancy could partially be attributed to the COO designation method used by Schmitz et al. [27], wherein classification was based on gene expression profiling. Assessment of *MYC* and *BCL2* translocation status within the validation set would be commendable; however, the translocation data were not publicly available.

Ennishi et al. recently reported that a double-hit gene signature found in 27% of GCB-DLBCL predicted inferior clinical outcomes irrespective of HGBL-DH/TH status [35]. Sha et al. performed gene expression profiling and identified a molecular high-grade group (so-called MHG) of DLBCL patients (9% of the cohort). Most of these patients had GCB-like DLBCL, half of whom had DH lymphoma [36]. Thus, further studies are required to elucidate the relationship between DE status, MHG signature, and clinical outcome in DE-DLBCL patients with the GCB phenotype.

This study harbors some limitations, stemming from being a retrospective study. The retrieval of the clinical information relied on the review of medical records, and thus is not entirely complete. Future studies with prospective assessment of risk factors in patients with DE-DLBCL should be warranted. In addition, failure to identify HGBL-DH/TH within DLBCL or HGBL by molecular analyses may have compromised precise clinicopathological analyses and prognostication in this study. 

In summary, we confirmed the prognostic value of MYC/BCL2 DE in DLBCL patients treated with R-CHOP irrespective of COO. We stratified DE-DLBCL patients into subgroups with different prognoses and found that DE-DLBCL patients with normal LDH levels had clinical outcomes similar to those of non-DE-DLBCL patients. These findings suggest that DE-DLBCL patients with elevated serum LDH levels may require more aggressive therapeutic interventions.

## 4. Materials and Methods 

### 4.1. Patients

A total of 461 consecutive adult (≥18 years old) patients newly diagnosed with aggressive B-cell lymphomas, including DLBCL (*n* = 417) and HGBL (*n* = 44), were enrolled in the study. Patients with primary mediastinal large B-cell lymphoma, primary central nervous system (CNS) lymphoma, Epstein–Barr virus (EBV)-positive DLBCL, and immunodeficiency-associated DLBCL, were excluded. All enrolled patients were diagnosed between 2014 and 2017 at the Seoul National University Hospital (SNUH) and had tumor tissues available for immunophenotyping. Clinical data and outcomes were evaluated by experienced hemato-oncologists blinded to the pathological data. OS was defined as the time between the date of initial diagnosis and the last follow-up or death from any cause. PFS was defined as the time between treatment initiation and tumor progression or relapse. The follow-up period ranged from 0 to 143.9 months (median, 22.7 months). As a validation cohort, we enrolled 260 adult patients newly diagnosed with DLBCL at the Seoul National University Bundang Hospital (SNUBH) between 2013 and 2018. The follow-up period of these patients ranged from 0 to 60.1 months (median, 15.5 months). This study was conducted in accordance with the recommendations of the World Medical Association Declaration of Helsinki and was approved by the Institutional Review Board (IRB) of SNUH (No.1506-080-681); informed consent for participation in the retrospective study was waived by the IRB. 

### 4.2. IHC and FISH 

IHC was prospectively performed on the routine diagnostic basis using representative whole formalin-fixed paraffin-embedded (FFPE) tissue sections and antibodies against CD3 (clone 2GV6; Ventana Medical Systems, Tucson, AZ, USA), CD20 (clone L26; DAKO, Carpinteria, CA, USA), BCL2 (clone 124; DAKO, Carpinteria, CA, USA), BCL6 (clone LN22; Novocastra, Newcastle, UK), CD10 (clone 56C6; Novocastra), MUM1 (clone Ma695; Novocastra), MYC (clone EP121; Cell Marque, Rocklin, CA, USA), and Ki-67 (clone MIB-1; DAKO). Staining was performed using a Ventana Benchmark XT (Ventana Medical Systems) or a Bond-Max autostainer (Leica Microsystems, Melbourne, Australia). COO was determined using the IHC-based Hans algorithm as previously described [29]. DE status was defined as the co-expression of MYC (in ≥40% of tumor cells) and BCL2 (in ≥70% of tumor cells) as previously described [12]. TMA was constructed using the FFPE tissue blocks from 50 selected cases with DLBCL, and we performed *MYC*, *BCL2*, *BCL6* FISH on the TMA. FISH was performed using Vysis LSI *BCL2* Dual Color Break Apart Rearrangement Probe (Vysis, Downers Grove, IL, USA), Vysis LSI *BCL6* Dual Color Break Apart Rearrangement Probe (Vysis), and Vysis LSI *MYC* Dual Color Break Apart Rearrangement Probe (Vysis)).

### 4.3. Validation Using Publicly Available Data

As a validation set, we used publicly available data generated by Schmitz et al. [27] and downloaded from the National Cancer Institute Genomic Data Commons (https://gdc.cancer.gov/about-data/publications/DLBCL-2018; acquired on 10 January 2020). Clinical data, including OS and PFS, were also obtained. BCL2 and MYC expression profiles from RNA sequencing gene expression data were matched with the clinical data. All 234 patients were treated with R-CHOP or CHOP-like chemotherapy, and gene expression values were presented as normalized fragments per kilobase per million (FPKM) values on a log2 scale. To identify the optimal cut-off values for BCL2 and MYC expression levels that more accurately stratified patients based on OS, we used the FPKM expression levels of BCL2 and MYC to define DE-DLBCL in this dataset. The R package “maxstat” was used to run maximally selected chi-square statistics [37]. The estimated cut-off values for BCL2 and MYC were 11.489 and 9.735, respectively. Patients with BCL2 and MYC expression levels higher than these cut-offs were defined as DE-DLBCL (*n* = 58). When OS and PFS were compared between DE-DLBCL and the others, significant differences were observed as expected (figure not shown; *p* = 0.007 and 0.001, respectively). 

### 4.4. Statistical Analysis 

All statistical analyses were performed using SPSS software (version 21; IBM Corp., Armonk, NY, USA) or R statistical package 3.6.0 (http://www.r-project.org). Variables were performed using Pearson’s χ^2^ test or Fisher’s exact test. Survival analysis was conducted using the Kaplan–Meier method with the log-rank test. Univariate and multivariate survival analyses were also performed using the Cox proportional hazards model. Two-sided *p*-values < 0.05 were considered statistically significant. 

## 5. Conclusions

MYC/BCL2 DE is a poor prognostic factor in DLBCL patients treated with R-CHOP irrespective of COO in the real world. DE-DLBCL patients could be stratified into subgroups with different prognoses and DE-DLBCL patients with normal LDH levels had clinical outcomes similar to those of non-DE-DLBCL patients, which suggests that DE-DLBCL patients with elevated LDH levels may require more aggressive therapeutic interventions.

## Figures and Tables

**Figure 1 cancers-12-03305-f001:**
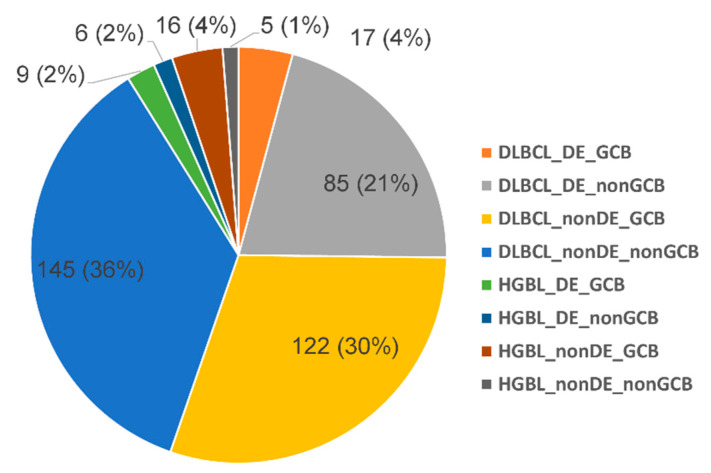
Immunophenotypic landscape of aggressive B-cell lymphoma. Stratification of aggressive B-cell lymphoma patients, including diffuse large B-cell lymphoma (DLBCL) and high-grade B-cell lymphoma (HGBL) based on MYC/BCL2 double expression (DE) status and cell-of-origin (COO).

**Figure 2 cancers-12-03305-f002:**
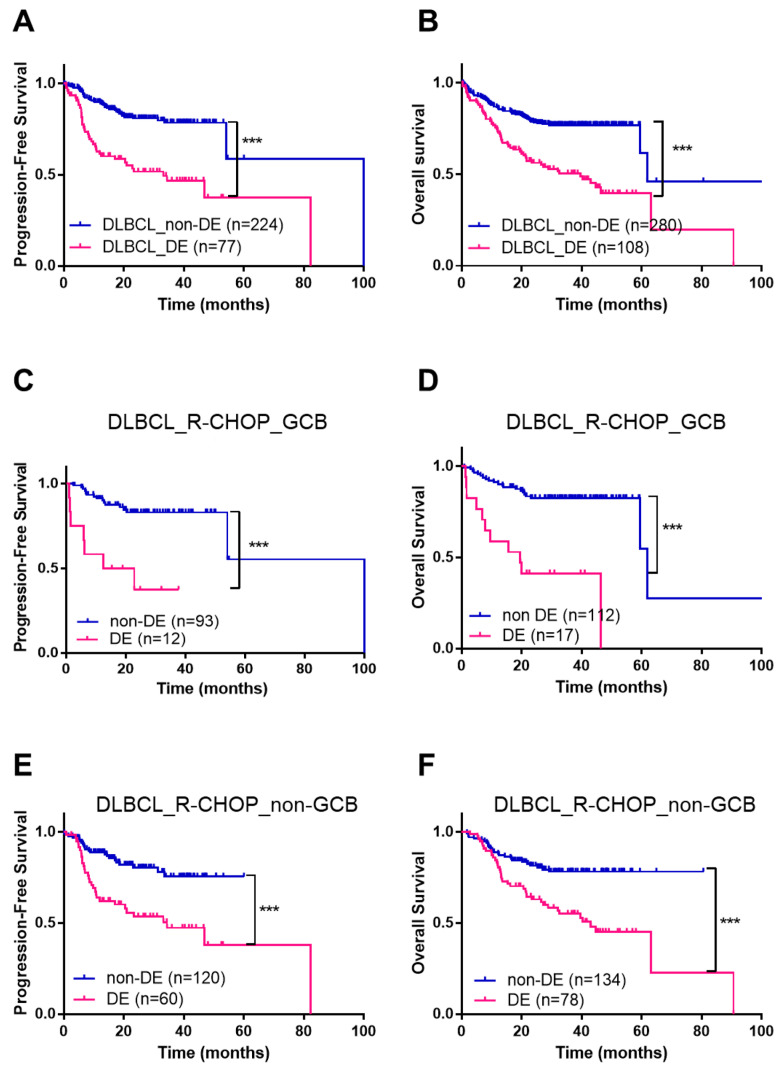
Effect of DE status and COO on the survival of DLBCL patients treated with R-CHOP. Kaplan-Meier curves showing PFS and OS of DLBCL patients according to the DE status (**A**,**B**). Kaplan-Meier curves showing PFS and OS in R-CHOP-treated DLBCL patients (GCB phenotype) according to the DE status (**C**,**D**). Kaplan-Meier curves showing PFS and OS of R-CHOP-treated DLBCL patients (non-GCB phenotype) according to the DE status (**E**,**F**). *** *p* < 0.001.

**Figure 3 cancers-12-03305-f003:**
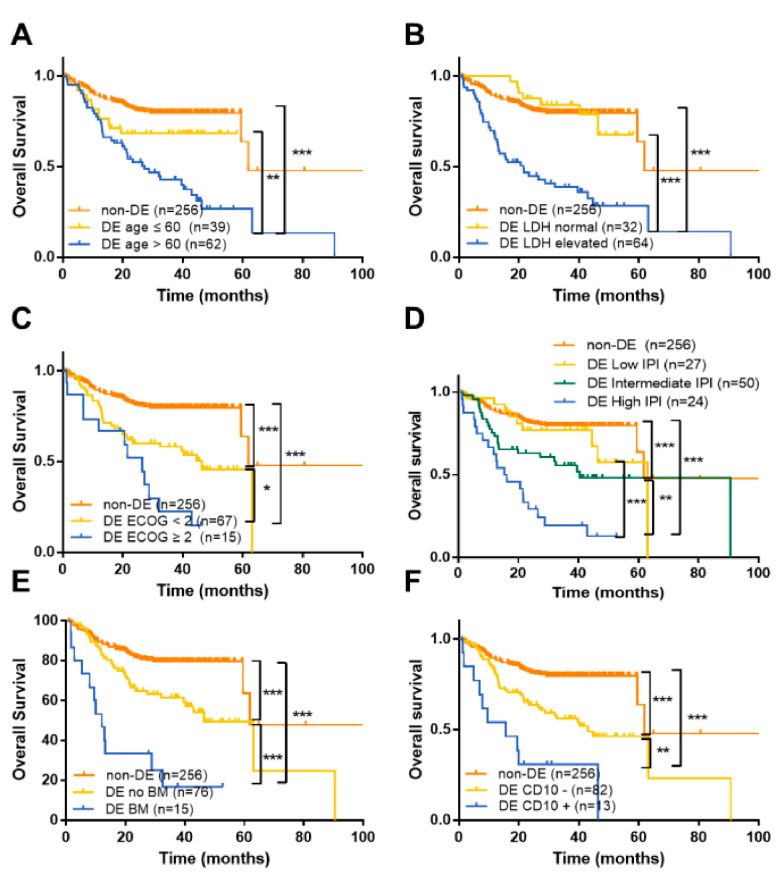
Survival of DLBCL patients treated with R-CHOP according to the DE status and clinicopathological parameters. OS of DLBCL patients treated with R-CHOP according to the DE status and age (**A**), serum LDH levels (**B**), ECOG PS (**C**), IPI score (**D**), BM involvement (**E**), CD10 expression (**F**), number of extranodal sites (**G**), and COO (**H**). * *p* < 0.05, ** *p* < 0.01, *** *p* < 0.001.

**Table 1 cancers-12-03305-t001:** Clinicopathological characteristics of patients with aggressive B-cell lymphoma.

Variables *		DLBCL (*n* = 417) No. (%)	HGBL (*n* = 44) No. (%)	*p*
Sex	male	242 (58.0)	31 (70.5)	0.146
	female	175 (42.0)	13 (29.5)	
Age, years	median ± SD	62 ± 14.094	56 ± 15.055	0.002
	mean ± SD	61.09 ± 14.094	54.18 ± 15.055	
Primary sites	nodal	191 (45.8)	18 (40.9)	0.633
	extranodal	226 (54.2)	26 (59.1)	
Ann Arbor stage	1	63 (15.4)	6 (14.3)	0.282
	2	100 (24.4)	8 (19.0)	
	3	72 (17.6)	4 (9.5)	
		174 (42.5)	24 (57.1)	
B symptoms	absent	381 (93.4)	39 (90.7)	0.522
	present	27 (6.6)	4 (9.3)	
Bulky disease ^†^	absent	375 (91.7)	27 (62.8)	<0.001
	present	34 (8.3)	16 (37.2)	
ECOG PS	0 or 1	288 (85.2)	33 (86.8)	1.000
	2 or more	50 (14.8)	5 (13.2)	
Serum LDH	normal	186 (47.6)	18 (43.9)	0.743
	elevated	205 (52.4)	23 (56.1)	
No. of extranodal sites	0 or 1	276 (69.0)	29 (69.0)	1.000
	2 or more	124 (31.0)	13 (31.0)	
BM involvement	absent	308 (84.8)	29 (72.5)	0.068
	present	55 (15.2)	11 (27.5)	
IPI	0-1 low risk	153 (37.0)	14 (32.6)	0.434
	2, low-int. risk	107 (25.8)	8 (18.6)	
	3, high-int. risk	86 (20.8)	13 (30.2)	
	4-5, high risk	68 (16.4)	8 (18.6)	
Tx regimen	R-CHOP	382 (97.9)	29 (69.0)	<0.001
	Others	8 (2.1)	13 (31.0)	
Response to Tx	CR	273 (86.9)	26 (81.3)	0.413
	non-CR	41 (13.1)	6 (18.8)	
BCL2	negative	191 (47.0)	20 (51.3)	0.619
	positive	215 (53.0)	19 (48.7)	
MYC	negative	227 (57.8)	10 (25.6)	<0.001
	positive	166 (42.2)	29 (74.4)	
MYC/BCL2 DE status	non-DE	280 (72.2)	21 (56.8)	0.058
	DE	108 (27.8)	16 (43.2)	
CD10	negative	289 (74.1)	16 (41.0)	<0.001
	positive	101 (25.9)	23 (59.0)	
BCL6	negative	177 (44.8)	9 (23.1)	0.01
	positive	218 (55.2)	30 (76.9)	
MUM1	negative	161 (41.1)	21 (56.8)	0.081
	positive	231 (58.9)	16 (43.2)	
COO	GCB	145 (37.3)	27 (69.2)	<0.001
	non-GCB	244 (62.7)	12 (30.8)	

***** Some variables have missing values. ^†^ Bulky disease was defined as tumor measured above 10 cm in the greatest dimension. DLBCL, diffuse large B-cell lymphoma; HGBL, high-grade B-cell lymphoma; ECOG PS, Eastern Cooperative Oncology Group Performance Status; LDH, lactate dehydrogenase; No., number; BM, bone marrow; IPI, International Prognostic Index; Int., intermediate; Tx, treatment; R-CHOP, rituximab, cyclophosphamide, doxorubicin, vincristine, and prednisone; CR, complete response; DE, double expression; COO, cell-of-origin; GCB, germinal center B-cell-like.

**Table 2 cancers-12-03305-t002:** The relationship between DE status and clinicopathological features of DLBCL patients.

Variables *		DLBCL		*p*
		Non-DE (*n* = 280) No. (%)	DE (*n* = 108) No. (%)	
Sex	male	169 (60.4)	58 (53.7)	0.251
	female	111 (39.6)	50 (46.3)	
Age, years	median ± SD	62 ± 14.60	64 ± 13.08	0.040
	mean ± SD	60.3 ± 14.60	63.6 ± 13.08	
Primary sites	nodal	126 (45.0)	52 (48.1)	0.649
	extranodal	154 (55.0)	56 (51.9)	
Ann Arbor stage	1	43 (15.6)	15 (14.2)	0.406
	2	69 (25.1)	21 (19.8)	
	3	49 (17.8)	16 (15.1)	
		114 (41.5)	54 (50.9)	
B symptoms	absent	256 (94.1)	97 (89.8)	0.182
	present	16 (5.9)	11 (10.2)	
Bulky disease	absent	249 (91.2)	101 (93.5)	0.538
	present	24 (8.8)	7 (6.5)	
ECOG PS	0 or 1	198 (86.8)	70 (80.5)	0.161
	2 or more	30 (13.2)	17 (19.5)	
Serum LDH	normal	139 (52.7)	35 (34.3)	0.002
	elevated	125 (47.3)	67 (65.7)	
No. of extranodal sites	0 or 1	185 (69.0)	69 (65.7)	0.539
	2 or more	83 (31.0)	36 (34.3)	
BM involvement	absent	208 (84.9)	79 (83.2)	0.739
	present	37 (15.1)	16 (16.8)	
IPI	0–1, low	110 (39.6)	30 (27.8)	0.042
	2, low-int.	72 (25.9)	26 (24.1)	
	3, high-int.	57 (20.5)	26 (24.1)	
	4–5, high	39 (14.0)	26 (24.1)	
Tx regimen	R-CHOP	256 (97.7)	101 (98.1)	1.000
	others	6 (2.3)	2 (1.9)	
Response to Tx	CR	196 (90.3)	63 (80.8)	0.042
	non-CR	21 (9.7)	15 (19.2)	
PD or relapse	no	183 (81.7)	38 (49.4)	<0.001
	yes	41 (18.3)	39 (50.6)	
BCL2	negative	179 (63.9)	0 (0)	<0.001
	positive	101 (36.1)	108 (100)	
MYC	negative	224 (80.0)	0 (0.0)	<0.001
	positive	56 (20.0)	108 (100)	
CD10	negative	184 (68.7)	89 (87.3)	<0.001
	positive	84 (31.3)	13 (12.7)	
BCL6	negative	115 (42.4)	46 (45.1)	0.641
	positive	156 (57.6)	56 (54.9)	
MUM1	negative	125 (46.5)	27 (26.5)	0.001
	positive	144 (53.5)	75 (73.5)	
COO	GCB	122 (45.7)	17 (16.7)	<0.001
	non-GCB	145 (54.3)	85 (83.3)	

* Some variables have missing values. Abbreviations: DLBCL, diffuse large B-cell lymphoma; DE, MYC and BCL2 double expression; ECOG PS, Eastern Cooperative Oncology Group Performance Status; LDH, lactate dehydrogenase; No., number; BM, bone marrow; IPI, International Prognostic Index; Int., intermediate; Tx, treatment; R-CHOP, rituximab, cyclophosphamide, doxorubicin, vincristine, and prednisone; CR, complete response; PD, progressive disease; COO, cell-of-origin; GCB, germinal center B-cell-like.

**Table 3 cancers-12-03305-t003:** Multivariate analysis of OS and PFS according to clinicopathological parameters in DLBCL patients treated with R-CHOP.

Variables	PFS			OS		
	HR	95% CI	*p*	HR	95% CI	*p*
Comparison with risk factors						
Age				1.033	1.014–1.053	0.001
Sex (female)				0.669	0.425–1.052	0.082
Ann Arbor Stage III/IV	2.436	1.290–4.599	0.006	2.064	1.195–3.565	0.009
ECOG PS of ≥2				2.278	1.407–3.687	0.001
Elevated serum LDH	2.610	1.424–4.783	0.002	4.522	2.456–8.328	<0.001
No. of extranodal sites ≥2						
MYC/BCL2 DE	2.885	1.707–4.876	<0.001	1.872	1.194–2.936	0.006
Comparison with IPI						
IPI score of ≥2	2.259	1.312–3.889	<0.001	3.641	2.101–6.312	<0.001
MYC/BCL2 DE	3.041	1.943–4.760	<0.001	2.678	1.819–3.941	<0.001

DLBCL, diffuse large B-cell lymphoma; PFS, progression-free survival; OS, overall survival; HR, hazard ratio; CI, confidence interval; ECOG PS, Eastern Cooperative Oncology Group Performance Status; LDH, lactate dehydrogenase; No., number; IPI, International Prognostic Index; R-CHOP, rituximab, cyclophosphamide, doxorubicin, vincristine, and prednisone; DE, double expression.

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
