# Peer review of "Immunophenotypic Landscape and Prognosis of Diffuse Large B-Cell Lymphoma with MYC/BCL2 Double Expression: An Analysis of A Prospectively Immunoprofiled Cohort"

_cancers, 2020, doi:10.3390/cancers12113305_

Round 1

Reviewer 1 Report

Han and colleagues here analyzed a cohort of 461 HGBLs patients prospectively and claimed a confirmation of a poor prognostic value of MYC/BCL2-DE in DLBCL irrespective COO and IPI index. Unfortunately, this speculation can be supported just by the univariate Cox survival analysis in GCB or non-GCB patients after R-CHOP treatment in Fig.2. I cannot see the same prognostic relevance from the multivariate analysis where the non-DE patients are not stratified for the additional risk factor considered. In particular LDH, age and IPI score seem to affect the difference between DE and not-DE patients. Actually the authors described how other risk factors are able to stratify DE-DLBCL, but in my opinion this is not much different from stratifying non-GCB patients on the basis of IPI score, since DE-DLBCL are mainly non-GCB.

The aim and the conclusions of the paper are a bit confusing and not well stated as well as statistical methods

Author Response

Reviewer 1

Han and colleagues here analyzed a cohort of 461 HGBLs patients prospectively and claimed a confirmation of a poor prognostic value of MYC/BCL2-DE in DLBCL irrespective COO and IPI index.

Unfortunately, this speculation can be supported just by the univariate Cox survival analysis in GCB or non-GCB patients after R-CHOP treatment in Fig.2. I cannot see the same prognostic relevance from the multivariate analysis where the non-DE patients are not stratified for the additional risk factor considered. In particular LDH, age and IPI score seem to affect the difference between DE and not-DE patients. Actually the authors described how other risk factors are able to stratify DE-DLBCL, but in my opinion this is not much different from stratifying non-GCB patients on the basis of IPI score, since DE-DLBCL are mainly non-GCB.

The aim and the conclusions of the paper are a bit confusing and not well stated as well as statistical methods

; We appreciate you for taking time and efforts to review our manuscript and for all the valuable comments. DE status was significantly associated with poor PFS and OS in patients with DLBCL treated with R-CHOP in multivariate analysis as shown in the Results section 2.4. (lines 167-171) and Table 3.

“Multivariate survival analysis revealed that DE status was a significant poor prognostic factor for PFS and OS, independently of age, sex, stage, ECOG PS, serum LDH level, and the number of extranodal sites (P < 0.001 for PFS and P = 0.006 for OS), as well as independently of IPI score (P < 0.001 for both PFS and OS; Table 3).”

Table 3. Multivariate analysis of OS and PFS according to clinicopathological parameters in DLBCL patients treated with R-CHOP

Variables

PFS

OS

HR

95% CI

P

HR

95% CI

P

Comparison with risk factors

Age

.

.

.

1.033

1.014-1.053

0.001

Sex (female)

.

.

.

0.669

0.425-1.052

0.082

Ann Arbor Stage III/IV

2.436

1.290-4.599

0.006

2.064

1.195-3.565

0.009

ECOG PS of ≥2

.

.

.

2.278

1.407-3.687

0.001

Elevated serum LDH

2.610

1.424-4.783

0.002

4.522

2.456-8.328

<0.001

No. of extranodal sites ≥2

.

.

.

.

.

.

MYC/BCL2 DE

2.885

1.707-4.876

<0.001

1.872

1.194-2.936

0.006

Comparison with IPI

IPI score of ≥2

2.259

1.312-3.889

<0.001

3.641

2.101-6.312

<0.001

MYC/BCL2 DE

3.041

1.943-4.760

<0.001

2.678

1.819-3.941

<0.001

Abbreviations: DLBCL, diffuse large B-cell lymphoma; PFS, progression-free survival; OS, overall survival; HR, hazard ratio; CI, confidence interval; ECOG PS, Eastern Cooperative Oncology Group Performance Status; LDH, lactate dehydrogenase; No., number; BM, bone marrow; IPI, International Prognostic Index; R-CHOP, rituximab, cyclophosphamide, doxorubicin, vincristine, and prednisone; COO, cell-of-origin; non-GCB, non-germinal center B-cell-like; DE, double expression

Actually, we did not include COO in the above multivariate analysis since immunohistochemically-defined COO had no prognostic significance in our cohort. However, we think that your concern that the prognostic impact of DE status might depend on the COO of DLBCL, is reasonable. Therefore, to address your concern, we carried out univariate and multivariate survival analyses within GCB- and non-GCB- DLBCLs, respectively; in brief, DE status predicted poor PFS and OS independently of IPI in both GCB and non-GCB DLBCL patients treated with R-CHOP. We described this finding in the manuscript and Supplementary Tables S2-3 as follows:

(lines 171-173)

“In addition, DE status was a significant poor prognostic factor for PFS and OS independently of IPI score in both GCB and non-GCB DLBCL groups, respectively (Supplementary Table S2 and S3).”

Supplementary Table S2. Multivariate survival analysis of OS and PFS according to clinicopathological parameters in GCB DLBCL patients treated with R-CHOP

Variables

PFS

OS

HR

95% CI

P

HR

95% CI

P

Comparison with risk factors

Ann Arbor Stage III-IV

6.007

2.031-17.767

0.001

3.707

1.213-11.324

0.021

ECOG PS 2

.

.

.

6.513

1.819-23.326

0.004

No. of extranodal sites 2

.

.

.

3.874

1.503-9.981

0.005

MYC/BCL2 DE status 

11.089

4.048-30.379

<0.001

9.814

3.342-28.823

<0.001

Comparison with IPI

IPI 2

6.08

2.059-17.952

0.001

8.093

3.246-20.173

<0.001

MYC/BCL2 DE status 

11.17

4.083-30.556

<0.001

12.952

5.024-33.391

<0.001

Abbreviations: DLBCL, diffuse large B-cell lymphoma; PFS, progression-free survival; OS, overall survival; HR, hazard ratio; CI, confidence interval; ECOG PS, Eastern Cooperative Oncology Group Performance Status; No., number; IPI, International Prognostic Index; R-CHOP, rituximab, cyclophosphamide, doxorubicin, vincristine, and prednisone; GCB, germinal center B-cell-like; DE, double expression

Supplementary Table S3. Multivariate survival analysis of OS and PFS according to clinicopathological parameters in non-GCB DLBCL patients treated with R-CHOP

Variables

PFS

OS

HR

95% CI

P

HR

95% CI

P

Comparison with risk factors

Age

.

.

.

2.062

1.039-4.091

0.038

Ann Arbor Stage III-IV

1.922

0.893-4.139

0.095

1.944

0.959-3.941

0.065

ECOG PS 2

.

.

.

2.255

1.268-4.011

0.006

Elevated serum LDH

4.360

1.909-9.957

<0.001

7.526

2.960-19.135

<0.001

MYC/BCL2 DE status 

1.899

1.003-3.594

0.049

.

.

.

Comparison with IPI

IPI 2

2.781

1.540-5.023

0.001

3.175

1.838-5.486

<0.001

MYC/BCL2 DE status 

2.224

1.264-3.985

0.006

2.235

1.323-3.776

0.003

Abbreviations: DLBCL, diffuse large B-cell lymphoma; PFS, progression-free survival; OS, overall survival; HR, hazard ratio; CI, confidence interval; ECOG PS, Eastern Cooperative Oncology Group Performance Status; No., number; IPI, International Prognostic Index; R-CHOP, rituximab, cyclophosphamide, doxorubicin, vincristine, and prednisone; non-GCB, non-germinal center B-cell-like; DE, double expression

Regarding the prognostic implication of LDH, age and IPI score, these were significantly associated with overall survival in patients with non-DE-DLBCL in the similar manner as in patients with DE-DLBCL.

In this study, we classified patients into three groups: non-DE-DLBCL, DE-DLBCL without a risk factor, and DE-DLBCL with a risk factor. We’d like to emphasize our findings that OS of DE-DLBCL patients who did not harbor risk factors (including older age, poor ECOG PS, BM involvement, involvement of two or more extranodal sites) were between the other two groups, i.e., non-DE-DLBCL patients and DE-DLBCL patients with these risk factors (P < 0.05 for every comparison, Figure 3). In contrast, DE-DLBCL patients with normal LDH levels exhibited similar PFS and OS to those of non-DE-DLBCL patients (Figure 3B and Supplementary Figure S3B). Thus, we propose that risk-stratification of DE-DLBCL patients based on LDH levels may guide clinical decision-making since these patients would follow the clinical course similar to those with non-DE DLBCL.

We again thank you for your valuable comments which helped us improve the integrity of our study. Any additional suggestions would be greatly appreciated if you find there is the need for further clarification.

Reviewer 2 Report

General - I think it would be helpful for transparency to include the number of samples for which FISH for MYC was done (really should have been all of these for an analysis like this) and what the results are.  As you know, most DHL is GC, so it is possible that the GC DE-DLBCL patients did worse here because of an overrepresentation of DHL patients who we know do badly with R-CHOP.

Line 56-59 ("...intensive treatment exhibit a better clinical outcome...") I would add the reference from Petrich, et al. Blood 2014 on this topic.

Lines 67-69 - "Nevertheless, debate remains over the most appropriate diagnostic approach for DH 68 lymphoma, as well as the value of immunohistochemical staining for MYC and BCL2 as a surrogate 69 clinical marker for fluorescence in situ hybridization (FISH) [1,19,20]"

There really is no debate re: how to diagnose DHL and IHC as surrogate for FISH.  The DHL criteria are well defined.  Also, IHC is clearly an inadequate surrogate for FISH because IHC defines DEL and FISH defines DHL.  Please remove this sentence.

Line 93 - please define bulky

LIne 143 - change the word prognosis to "overall survival"

Author Response

Authors’ Response Letter to the Reviewers’ Comments

Reviewer 2

General - I think it would be helpful for transparency to include the number of samples for which FISH for MYC was done (really should have been all of these for an analysis like this) and what the results are.

; In the revised manuscript, we included the number of samples tested for FISH analyses, as well as the results and relevant methods, as follows:

(lines 273-280)

“To address this issue, we retrospectively performed tissue microarray (TMA)-based fluorescent in situ hybridization (FISH) analyses in 50 selected DLBCLs with available tissue samples for TMA construction. However, as many DE-DLBCL patients with GCB phenotype were diagnosed based on biopsies, and old FFPE specimens failed to produce robust signals. 28% and 16% of the tested cases for MYC and BCL6, respectively, resulted in inadequate signals, and BCL2 FISH did not produce interpretable signals in all samples, hindering the identification of HGBL-DH/TH within this sub-cohort."

(lines 332-336)

“TMA was constructed using the FFPE tissue blocks from 50 selected cases with DLBCL, and we performed MYC, BCL2, BCL6 FISH on the TMA. FISH was performed using Vysis LSI BCL2 Dual Color Break Apart Rearrangement Probe (Vysis, Downers Grove, IL, USA, Vysis LSI BCL6 Dual Color, Break Apart Rearrangement Probe (Vysis) and Vysis LSI MYC Dual Color Break Apart Rearrangement Probe (Vysis) “

As you know, most DHL is GC, so it is possible that the GC DE-DLBCL patients did worse here because of an overrepresentation of DHL patients who we know do badly with R-CHOP.

; We fully agree with your insightful comments. We acknowledge the possible confounding effect of DHL within GC DE-DLBCLs, therefore, we attempted to determine the DH or TH status of the study population by FISH analyses. However, FISH analyses on the old, FFPE archived tissue failed to produce interpretable signals in substantial portion of the cases. We regret not being able to resolve this confounding issue, and fully addressed this as limitation in the Discussion section of the revised manuscript as follows:

(lines 296-297)

“In addition, failure of identifying HGBL-DH/TH within DLBCL or HGBL by molecular analyses may have compromised precise clinicopathological analyses and prognostication in this study.”

In addition, DLBCLs with “MHG” and “double-hit signature” are known to be associated with poor clinical outcome irrespective of DH/TH status. Patients within these categories could have been included in our study population, representing the worst prognostic group of DE-DLBCL patients. Therefore, further studies comprising molecular techniques including gene expression profiling would be required to investigate the heterogeneity within DE-DLBCLs, and we thoroughly discussed these unresolved issues in the Discussion section as follows:

(lines 287-292)

“Ennishi et al. recently reported that a double-hit gene signature found in 27% of GCB-DLBCL predicted inferior clinical outcomes irrespective of HGBL-DH/TH status [36]. Sha et al. performed gene expression profiling and identified a molecular high-grade group (so-called MHG) DLBCL patients (9% of the cohort). Most of these patients had GCB-like DLBCL, half of whom had DH lymphoma [37]. Thus, further studies are required to elucidate the relationship between DE status, MHG signature, and clinical outcome in DE-DLBCL patients with GCB phenotype.”

Line 56-59 ("...intensive treatment exhibit a better clinical outcome...") I would add the reference from Petrich, et al. Blood 2014 on this topic.

; We cited the article following the reviewer’s suggestion, as follows:

(lines 60-62)

“Evidence from retrospective studies shows that DH lymphoma patients receiving intensive treatment exhibit a better clinical outcome than patients treated with rituximab plus cyclophosphamide, doxorubicin, vincristine, and prednisone (R-CHOP) [6-8].”

(lines 416-420)

“8.        Petrich, A.M.; Gandhi, M.; Jovanovic, B.; Castillo, J.J.; Rajguru, S.; Yang, D.T.; Shah, K.A.; Whyman, J.D.; Lansigan, F.; Hernandez-Ilizaliturri, F.J., et al. Impact of induction regimen and stem cell transplantation on outcomes in double-hit lymphoma: a multicenter retrospective analysis. Blood 2014, 124, 2354-2361, doi:10.1182/blood-2014-05-578963.“

Lines 67-69 - "Nevertheless, debate remains over the most appropriate diagnostic approach for DH lymphoma, as well as the value of immunohistochemical staining for MYC and BCL2 as a surrogate clinical marker for fluorescence in situ hybridization (FISH) [1,19,20]"

There really is no debate re: how to diagnose DHL and IHC as surrogate for FISH. The DHL criteria are well defined. Also, IHC is clearly an inadequate surrogate for FISH because IHC defines DEL and FISH defines DHL. Please remove this sentence.

; We removed the sentence following the reviewer’s suggestion.

Line 93 - please define bulky

; Bulky disease was defined as tumor measured above 10 cm in the greatest dimension. We added this definition at the footnote of the Table 1, as follows:

Bulky disease was defined as tumor measured above 10 cm in the greatest dimension.”

Line 143 - change the word prognosis to "overall survival"

; Changes were made following the reviewer’s suggestion, as follows (lines 217-219):

“There was no difference in the overall survival (OS) of patients with DLBCLs and HGBLs in the whole cohort (Supplementary Figure S2A) or in the sub-cohort of patients treated with R-CHOP (Supplementary Figure S2B).”

Reviewer 3 Report

DLBCL with MYC/BCL2 protein coexpression (DEL) represents a poor prognostic group. This is a single-center and large cohort study including 417 cases of DLBCL and 44 cases of high grade B-cell lymphoma (HGBL).  DE is observed in 27.8% of DLBCL and 43.2% of HGBL. This study confirms the previous findings that DE is significantly associated with non-GC type and predicts poor PFS and OS in patients with R-CHOP-treated DLBCL. The findings also suggest that risk-stratification of DE-DLBCL patients based on LDH levels may guide clinical decision-making for DE-DLBCL patients.

Comments to the author:

  1. DLBCL, Burkitt lymphoma, and high grade B cell lymphoma (HGBL) are aggressive B-cell lymphomas. The diagnostic entity “B-cell lymphoma, unclassifiable, with features intermediate between DLBCL and Burkitt lymphoma” has been mostly replaced by “HGBL with MYC and BCL2 and/or BCL6 rearrangements” and “HGBL, NOS” in the 2016 WHO classification. “BLL” is not a recognized entity in the revised WHO classification and many of these cases are classified as HGBL, Burkitt-like lymphoma with 11q aberration or Burkitt lymphoma.  Therefore, the “BLL” cases in this cohort need to be specified according to the WHO classification.

  1. I think that it would be better to focus on DLBCL only.

  1. Clinical information is incomplete in general. For instance, clinical outcomes are missing in a large portion of the cases with Tx information.

  1. Where are the data for the 260 validation cohort cases?

  1. Figure 1: delete 1B.

Author Response

Authors’ Response Letter to the Reviewers’ Comments

Reviewer 3

DLBCL with MYC/BCL2 protein coexpression (DEL) represents a poor prognostic group. This is a single-center and large cohort study including 417 cases of DLBCL and 44 cases of high grade B-cell lymphoma (HGBL). DE is observed in 27.8% of DLBCL and 43.2% of HGBL. This study confirms the previous findings that DE is significantly associated with non-GC type and predicts poor PFS and OS in patients with R-CHOP-treated DLBCL. The findings also suggest that risk-stratification of DE-DLBCL patients based on LDH levels may guide clinical decision-making for DE-DLBCL patients.

  1. DLBCL, Burkitt lymphoma, and high grade B cell lymphoma (HGBL) are aggressive B-cell lymphomas. The diagnostic entity “B-cell lymphoma, unclassifiable, with features intermediate between DLBCL and Burkitt lymphoma” has been mostly replaced by “HGBL with MYC and BCL2 and/or BCL6 rearrangements” and “HGBL, NOS” in the 2016 WHO classification. “BLL” is not a recognized entity in the revised WHO classification and many of these cases are classified as HGBL, Burkitt-like lymphoma with 11q aberration or Burkitt lymphoma. Therefore, the “BLL” cases in this cohort need to be specified according to the WHO classification.

; We agree with the reviewer that some of the terminologies were not based on the current WHO classification; the terms “HGBL” and “BLL” used in our submitted manuscript (cancers-945157) refer to “aggressive B-cell lymphoma” and “high-grade B-cell lymphoma (HGBL)”, respectively, according to the revised 2017 WHO classification. In our revised manuscript (cancers-945157R1), we re-named the disease entities in line with the revised 2017 WHO classification and made relevant changes throughout the manuscript. Briefly, we established a cohort of aggressive B-cell lymphoma, composed of 417 patients with diffuse large B cell lymphoma (DLBCL) and 44 patients with high-grade B-cell lymphoma (HGBL).

  1. I think that it would be better to focus on DLBCL only.

; We thank the reviewer for the helpful comment. The reason that our study population encompassed the patients with HGBL in addition to those with DLBCL is to provide a better immunophenotypic picture of these entities. The majority of the studies on clinicopathological features of aggressive B-cell lymphomas were performed on the Western countries, further complicating the differences on the epidemiology of lymphomas in Asian ethnicity and Western population. Therefore, clinicopathological data of Asian patients with aggressive B-cell lymphoma is underrepresented. For this reason, we constructed a large cohort of Korean patients with aggressive B-cell lymphoma along with prospectively assessed immunophenotypic data.

While revising our manuscript, we found that the transition between the part focusing on the immunophenotypic features of DLBCL/HGBL and the part on the prognostic stratification of DE-DLBCL is not clearly explained, thus difficult to follow and confusing. Therefore, we re-wrote the parts of Results section to deliver our message more efficiently.

In addition, considering your comment, we sought to emphasize the key points of our study more efficiently. We changed the title of manuscript from “Immunophenotypic landscape of high-grade B-cell lymphomas and prognosis of diffuse large B-cell lymphoma with MYC/BCL2 double expression: an analysis of a prospectively immunoprofiled cohort” to “Immunophenotypic landscape and prognosis of diffuse large B-cell lymphoma with MYC/BCL2 double expression: an analysis of a prospectively immunoprofiled cohort”, to be more focused on the DLBCLs. Besides, we moved the panels of survival analyses including HGBL patients from Figure 2 into Supplementary Figures S2A-B, and left the prognostic stratification within DLBCLs on Figure 2, to help readers to concentrate on the major findings regarding DLBCL.

[Please see detached PDF file of this response for relating image]

  1. Clinical information is incomplete in general. For instance, clinical outcomes are missing in a large portion of the cases with Tx information.

; Since this study is a retrospective study, retrieval of the clinical information was largely based on the review of medical records. Despite, we managed to gather treatment response records from the substantial portion of the study population: 75.3 % (314/417) and 72.7% (32/44) from the patients with DLBCL and HGBL. In addition, since the immunohistochemical study was performed prospectively in this consecutive population as routine diagnostic procedure, DE status was available in almost all cases – 93.8% (391/417) and 88.1% (37/42) of DLBCLs and HGBLs – in addition to minimalizing selection bias. Therefore, although not entirely complete, our clinicopathological analyses provide useful information regarding the course of DE-DLBCLs according to the risk factors. Indeed, we fully addressed this issue in the Discussion section of the revised manuscript as follows:

(lines 293-295)

“This study harbors some limitations, stemming from being a retrospective study. The retrieval of the clinical information relied on the review of medical records, thus not entirely complete. Future studies with prospective assessment of risk factors in patients with DE-DLBCL should be warranted.”

  1. Where are the data for the 260 validation cohort cases?

; We regret not including the detailed information on the validation cohort from another hospital. We provided the clinicopathological data for 260 patients in the validation cohort in the Supplementary Table S4 of the revised manuscript.

Supplementary Table S4. Clinicopathological characteristics of patients in the validation set from SNUBH.

Variables*

DLBCL (n=260)

No. (%)

Sex

male

160 (61.5)

female

100 (38.5)

Age, years

median ± SD

65.50 ± 16.104

mean ± SD

62.25 ± 16.104

Primary sites

nodal

102 (39.2)

extranodal

158 (60.8)

Ann Arbor stage

1 or 2

101 (39.9)

3 or 4

152 (60.1)

B symptoms

absent

69 (87.3)

present

10 (12.7)

Bulky disease

absent

198 (77.3)

present

58 (22.7)

ECOG PS

0 or 1

9 (47.4)

2 or more

10 (52.6)

Serum LDH

normal

200 (76.9)

elevated

60 (23.1)

No. of extranodal sites

0 or 1

196 (76.9)

2 or more

59 (23.1)

BM involvement

absent

165 (80.1)

present

41 (19.9)

IPI

0-1, low risk

65 (80.3)

2, low-int risk

3 (3.7)

3, high-int risk

5 (6.2)

4-5, high risk

8 (9.9)

Tx regimen

R-CHOP

225 (93.4)

others

16 (6.6)

Response to Tx

CR

136 (69.7)

non-CR

59 (30.3)

BCL2

negative

72 (27.7)

positive

188 (72.3)

MYC

negative

180 (69.2)

positive

80 (30.8)

MYC/BCL2 DE status 

Non-DE

200 (76.9)

DE

60 (23.1)

CD10

negative

200 (76.9)

positive

60 (23.1)

BCL6

negative

200 (76.9)

positive

60 (23.1)

MUM1

negative

200 (77.5)

positive

58 (22.5)

COO

GCB

82 (31.5)

non-GCB

178 (68.5)

*Some variables have missing values.

Bulky disease was defined as tumor measured above 10 cm in the greatest dimension.

Abbreviations: DLBCL, diffuse large B-cell lymphoma; ECOG PS, Eastern Cooperative Oncology Group Performance Status; LDH, lactate dehydrogenase; No., number; BM, bone marrow; IPI, International Prognostic Index; Int., intermediate; Tx, treatment; R-CHOP, rituximab, cyclophosphamide, doxorubicin, vincristine, and prednisone; CR, complete response; DE, double expression; COO, cell-of-origin; GCB, germinal center B-cell-like

Furthermore, we described some interesting results found in this validation cohort in the Results section and Supplementary Figure S6, as follows:

(lines 268-270)

“Hans algorithm-based COO analysis in a validation cohort from SNUBH (Supplementary Table S4) confirmed that CD10 expression and GCB phenotype were associated with shorter survival in DE-DLBCL patients (Supplementary Figure S7).”

[Please see detached PDF file of this response for relating image]

  1. Figure 1: delete 1B.

; We deleted the Figure 1B and made relevant changes throughout the manuscript.

Round 2

Reviewer 1 Report

The authors substantially improved their study an fulfilled my requests. I suggest the paper for publication in present form

Reviewer 3 Report

No additional comments,